# SAM2Long: Enhancing SAM2 for Long Video Segmentation with a Training-Free Memory Tree

## Abstract

The Segment Anything Model 2 (SAM2) has emerged as a powerful foundation model for object segmentation in both images and videos, paving the way for various downstream video applications. The crucial design of SAM2 for video segmentation is its memory module, which prompts object-aware memories from previous frames for current frame prediction. However, its greedy-selection memory design suffers from the "error accumulation" problem, where an errored or missed mask will cascade and influence the segmentation of the subsequent frames, which limits the performance of SAM2 toward complex long-term videos. To this end, we introduce SAM2Long, an improved **training-free** video object segmentation strategy, which considers the segmentation uncertainty within each frame and chooses the video-level optimal results from multiple segmentation pathways in a constrained tree search manner. In practice, we maintain a fixed number of segmentation pathways throughout the video. For each frame, multiple masks are proposed based on the existing pathways, creating various candidate branches. We then select the same fixed number of branches with higher cumulative scores as the new pathways for the next frame. After processing the final frame, the pathway with the highest cumulative score is chosen as the final segmentation result. Benefiting from its heuristic search design, SAM2Long is robust toward occlusions and object reappearances, and can effectively segment and track objects for complex long-term videos. Without introducing any additional parameters or further training, SAM2Long significantly outperforms SAM2 on six VOS benchmarks. Notably, it achieves an average improvement of 3.8 points across all model sizes and, in some cases, up to 5 points in $\mathcal{J}\&\mathcal{F}$ on long-term video object segmentation benchmarks SA-V and LVOS.

## 1 Introduction

The Segment Anything Model 2 (SAM2) has gained significant attention as a unified foundational model for promptable object segmentation in both images and videos. Notably, SAM2 (Ravi et al., 2024) has achieved state-of-the-art performance across various video object segmentation tasks, significantly surpassing previous methods. Building upon the original SAM (Kirillov et al., 2023), SAM2 incorporates a memory module that enables it to generate masklet predictions using stored memory contexts from previously observed frames. This module allows SAM2 to seamlessly extend SAM into the video domain, processing video frames sequentially, attending to the prior memories of the target object, and maintaining object coherence over time.

While SAM2 demonstrates strong performance in video segmentation, its greedy segmentation strategy struggles to handle complex video scenarios with frequent occlusions and object reappearance. In detail, SAM2 confidently and accurately segments frames when clear visual cues are present. However, in scenarios with occlusions or reappearing objects, it can produce mask proposals that are highly variable and uncertain. Regardless of the frame's complexity, a uniform greedy selection strategy is applied to both scenarios: the mask with the highest predicted IoU is selected. Such greedy choice works well for the easy cases but raises the error potential for the challenging frames. Once an incorrect mask is selected into memory, it is uncorrectable and will mislead the segmentation of the subsequent frames. We show such an "error accumulation" problem in Figure 1 both qualitatively and

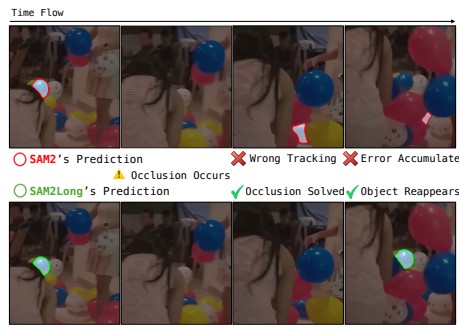

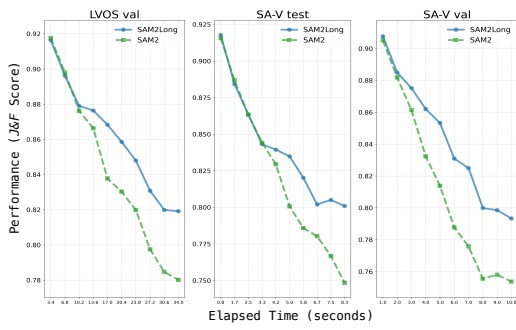

(a) Comparison in handling object occlusion over time. (b) Per-frame performance comparison across three benchmarks.

Figure 1: Comparison of occlusion handling and long-term compatibility between SAM2 and SAM2Long. (a) When an occlusion occurs, SAM2 may lose track or follow the wrong object, leading to accumulated errors. In contrast, SAM2Long utilizes memory tree search to recover when the object reappears. (b) The per-frame $\mathcal{J}\&\mathcal{F}$ scores of the predicted masks are plotted at specific timestamps on the LVOS and SA-V datasets. SAM2Long demonstrates greater resilience to elapsed time compared to SAM2, maintaining superior performance over longer periods.

quantitatively. The performance of SAM2 progressively deteriorates as the propagation extends into the later temporal segment, highlighting its limitations in maintaining accurate tracking over time.

To this end, we redesign the memory module of SAM2 to enhance its long-term compatibility and robustness against occlusions and error propagation. Our improvement is completely free of additional training and does not introduce any external parameters, but simply unleashes the potential of SAM2 itself. Our approach is motivated by the observation that the SAM2 mask decoder generates multiple diverse masks, accompanied by predicted IoU scores and an occlusion score when handling challenging and ambiguous cases. However, SAM2 only selects a single mask as memory, sometimes disregarding the correct one. To address this, we aim to equip SAM2 with multiple memory pathways, allowing various masks to be stored as memory at each time step, thereby improving predictions for subsequent frames.

In particular, we introduce a novel constrained tree memory structure, which maintains a fixed number of memory pathways over time to explore multiple segmentation hypotheses with efficiently managed computational resources. At each time step, based on a set of memory pathways, each with its own memory bank and cumulative score (accumulated logarithm of the predicted IoU scores across the pathway), we produce multiple candidate branches for the current frame. Then, among all the branches, we select the same fixed number of branches with higher cumulative scores and prune other branches, thereby constraining the tree's growth. After processing the final frame, the pathway with the highest cumulative score is selected as the final segmentation result. Moreover, to prevent premature convergence on incorrect predictions, we select hypotheses with distinct predicted masks when their occlusion scores indicate uncertainty, in order to maintain diversity in the tree branches. This tree-like memory structure augments SAM2's ability to effectively overcome error accumulation.

Within each pathway, we construct an object-aware memory bank that selectively includes frames with confidently detected objects and high-quality segmentation masks, based on the predicted occlusion scores and IoU scores. Instead of simply storing the nearest frames as SAM2 does, we filter out frames where the object may be occluded or poorly segmented. This ensures that the memory bank provides effective object cues for the current frame's segmentation. Additionally, we modulate the memory attention calculation by weighting memory entries according to their occlusion scores, emphasizing more reliable entries during cross-attention. These strategies help SAM2 focus on reliable object clues and improve segmentation accuracy with negligible computational overhead. As evidenced in Figure 1(a), our approach successfully resolves occlusions and re-tracks the recurring balloon, where SAM2 fails.

We provide a comprehensive evaluation demonstrating that SAM2Long consistently outperforms SAM2 across six VOS benchmarks, particularly excelling in long-term and occlusion-heavy scenarios. For instance, on the challenging SA-V validation set, SAM2Long-Large improves the $\mathcal{J}\&\mathcal{F}$ score

by 4.7 points, and SAM2Long-Small shows an impressive 4.9-point gain over the same size SAM2 model. Similar trends are observed on the LVOS validation set, where SAM2Long-Large surpasses SAM2-Large by 4.2 points. These consistent improvements, which range from 2.9 to 4.9 points across different model sizes, clearly indicate the effectiveness of our proposed method. Furthermore, as illustrated in Figure 1(b), the per-frame performance gap between SAM2Long and SAM2 widens over time, showcasing SAM2Long excels in long-term tracking scenarios. With these results, we believe SAM2Long sets a new standard for video object segmentation based on SAM2 in complex, real-world applications, delivering superior performance without any additional training or external parameters.

## 2 RELATED WORK

### 2.1 VIDEO OBJECT SEGMENTATION

Perceiving the environment in terms of objects is a fundamental cognitive ability of humans. In computer vision, Video Object Segmentation (VOS) tasks aim to replicate this capability by requiring models to segment and track specified objects within video sequences. A substantial amount of research has been conducted on video object segmentation in recent decades (Fan et al., 2019; Oh et al., 2019; Hu et al., 2018a; Oh et al., 2018; Perazzi et al., 2017; Wang et al., 2019; Hu et al., 2018b; Li & Loy, 2018; Bao et al., 2018; Zhang et al., 2019; Li et al., 2020; Johnander et al., 2019; Zhang et al., 2023; Ventura et al., 2019; Li et al., 2022; Wu et al., 2023; Wang et al., 2023).

There are two main protocols for evaluating VOS models (Pont-Tuset et al., 2017; Perazzi et al., 2016): semi-supervised and unsupervised video object segmentation. In semi-supervised VOS, the first-frame mask of the objects of interest is provided, and the model tracks these objects in subsequent frames. In unsupervised VOS, the model directly segments the most salient objects from the background without any reference. It is important to note that these protocols are defined in the inference phase, and VOS methods can leverage ground truth annotations during the training stage.

In this paper, we explore SAM2 (Ravi et al., 2024), for its application in semi-supervised VOS. We enhance the memory design of SAM2, significantly improving mask propagation performance without requiring any additional training.

### 2.2 MEMORY-BASED VOS

Video object segmentation remains an unsolved challenge due to the inherent complexity of video scenes. Objects in videos can undergo deformation (Tokmakov et al., 2023), exhibit dynamic motion (Brox & Malik, 2010), reappear over long durations (Hong et al., 2024; 2023), and experience occlusion (Ding et al., 2023), among other challenges. To address the above challenges, adopting a memory architecture to store the object information from past frames is indispensable for accurately tracking objects in video. Previous methods (Bhat et al., 2020; Caelles et al., 2017; Maninis et al., 2018; Robinson et al., 2020; Voigtlaender & Leibe, 2017) treat VOS as an online learning task, where networks are test-time tuned on the first-frame annotation. However, this approach was time-consuming due to test-time fine-tuning. Other techniques (Chen et al., 2018; Hu et al., 2018b; Voigtlaender et al., 2019; Yang et al., 2018; 2020; 2021b) use template matching, but they lack the capability of tracking under occlusion.

More recent approaches have introduced efficient memory reading mechanisms, utilizing either pixel-level attention (Cheng et al., 2023; Zhou et al., 2024; Duke et al., 2021; Liang et al., 2020; Oh et al., 2018; Seong et al., 2020; Cheng & Schwing, 2022; Xie et al., 2021; Yang & Yang, 2022; Yang et al., 2021a) or object-level attention (Athar et al., 2023; 2022; Cheng et al., 2024). A prominent example is XMem (Cheng & Schwing, 2022), which leverages a hierarchical memory structure for pixel-level memory reading combined. Building on XMem's framework, Cutie (Cheng et al., 2024) further improves segmentation accuracy by processing pixel features at the object level to better handle complex scenarios.

The latest SAM2 (Ravi et al., 2024) incorporates a simple memory module on top of the image-based SAM (Kirillov et al., 2023), enabling it to function for VOS tasks. However, by selecting only the temporally nearest frames as memory, SAM2 struggles with challenging cases involving long-term

reappearing objects and confusingly similar objects. we redesign SAM2's memory to maintain multiple potential correct masks, making the model more object-aware and robust.

## 2.3 SEGMENT ANYTHING MODEL

Segment Anything Model (SAM) (Kirillov et al., 2023) is recognized as a milestone vision foundation model that can segment any object in an image using interactive prompts. Its impressive zero-shot transfer performance has shown great versatility in various vision tasks, including segmentation applications (Li et al., 2023; Ma et al., 2024; Xu et al., 2024), image editing (Gao et al., 2023) and object reconstruction (Lin et al., 2024).

Building on SAM, SAM 2 (Ravi et al., 2024) extends its functionality to video segmentation through a memory-based transformer architecture for real-time video processing. SAM 2's memory stores information about objects and past interactions, enabling it to generate segmentation masks across video frames more accurately and efficiently than previous methods. To further enhance SAM 2, we introduce a constrained memory tree structure. This training-free design leverages the SAM2's ability to generate multiple candidate mask proposals with predicted IoU and occlusion score, mitigating error accumulation during segmentation.

## 3 METHOD

### 3.1 PRELIMINARY ON SAM2

SAM2 (Ravi et al., 2024) begins with an image encoder that encodes each input frame into embeddings. In contrast to SAM, where frame embeddings are fed directly into the mask decoder, SAM2 incorporates a memory module that conditions the current frame's features on both previous and prompted frames. Specifically, for semi-supervised video object segmentation tasks, SAM2 maintains a memory bank at each time step $t \geq 1$:

$$\mathcal{M}_t = \left\{ \mathbf{M}_\tau \in \mathbb{R}^{K \times C} \right\}_{\tau \in \mathcal{I}},$$

where $K$ is the number of memory tokens per frame, $C$ is the channel dimension, and $\mathcal{I}$ is the set of frame indices included in the memory. In SAM2, memory set $\mathcal{I}$ stores up to $N$ of the most recent frames, along with the initial mask, using a First-In-First-Out (FIFO) queue mechanism.

Each memory entry consists of two components: (1) the spatial embedding fused with the predicted mask (generated by the memory encoder), and (2) the object-level pointer (generated by the mask decoder). After cross-attending to the memory, the current frame's features integrate both fine-grained correspondences and object-level semantic information.[1]

The mask decoder, which is lightweight and retains the efficiency of SAM, then generates three predicted masks for the current frame. Each mask is accompanied by a predicted Intersection over Union (IoU) score $\text{IoU}_t \geq 0$ and an output mask token. Additionally, the mask decoder predicts a single occlusion score $o_t$ for the frame, where $o_t > 0$ indicates object presence, $o_t < 0$ indicates absence, and the absolute value $|o_t|$ depicts the model's confidence. The mask with the highest predicted IoU score is selected as the final prediction, and its corresponding output token is transformed into the object pointer for use as the memory.

### 3.2 CONSTRAINED TREE MEMORY WITH UNCERTAINTY HANDLING

To enhance SAM2's robustness towards long-term and ambiguous cases, we propose a constrained tree memory structure that enables the model to explore various object states over time with minimal computational overhead. We show the high-level pipeline in Figure 2. This tree-based approach maintains multiple plausible pathways and mitigates the effects of occlusions and erroneous predictions.

Specifically, at each time step $t$, we maintain a set of $P$ memory pathways, each with a memory bank $\mathcal{M}_t^p$ and a cumulative score $S_p[t]$, representing a possible segmentation hypothesis up to frame $t$. Conditioned on the memory bank of each pathway $p$, the SAM2 decoder head generates three mask

---

[1]In practice, SAM2 stores more object pointers than spatial embeddings, as pointers are lighter. We assume equal numbers of both components solely for illustrative purposes, without altering the actual implementation.

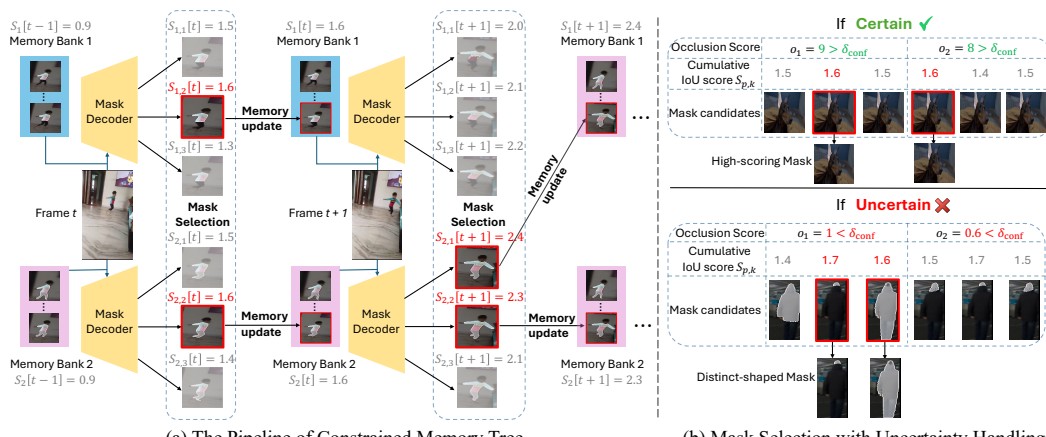

(a) The Pipeline of Constrained Memory Tree      (b) Mask Selection with Uncertainty Handling

Figure 2: (a) The pipeline of constrained memory tree: At each time step $t$, we maintain multiple memory pathways, each containing a memory bank and a cumulative score $S_p[t]$. The input frame is processed through the mask decoder conditioned on the memory bank, generating three mask candidates for each pathway. The candidates with the highest updated cumulative scores $S_{p,k}[t]$ are carried forward to the next time step. (b) Mask selection with uncertainty handling: When the maximum absolute occlusion score exceeds the threshold $\delta_{\text{conf}}$ (Certain), the high-scoring mask is selected. Otherwise (Uncertain), distinct mask candidates are picked to avoid incorrect convergence.

candidates along with their predicted IoU scores, denoted as $\text{IoU}_t^{p,1}$, $\text{IoU}_t^{p,2}$, and $\text{IoU}_t^{p,3}$. This process expands the tree by branching each existing pathway into three new candidates. As a result, there are a total of $3P$ possible pathways at each time step. We then calculate the cumulative scores for each possible pathway by adding the logarithm of its IoU score to the pathway's previous score:

$$S_{p,k}[t] = S_p[t-1] + \log(\text{IoU}_t^{p,k} + \epsilon), \quad \text{for } k = 1, 2, 3,$$

where $\epsilon$ is a small constant to prevent the logarithm of zero.

However, continuously tripling the pathways would lead to unacceptable computational and memory costs. Therefore, to manage computational complexity and memory usage, we implement a pruning strategy that selects the top $P$ pathways with the highest cumulative scores to carry forward to the next time step. This selection not only retains the most promising segmentation hypotheses but also constrains the tree-based memory, ensuring computational efficiency. Finally, we output the segmentation pathway with the highest cumulative score as the ultimate result.

Compared to SAM2, our approach introduces additional computation mainly by increasing the number of passes through the mask decoder and memory module. Notably, these components are lightweight relative to the image encoder. For instance, the image encoder of SAM2-Large consists of 212M parameters while the total parameter of SAM2-Large is 224M. Since we process the image encoder only once just as SAM2 does, the introduction of a memory tree adds negligible computational cost while significantly enhancing SAM2's robustness against error-prone cases.

**Uncertainty Handling.** Unfortunately, there are times when all pathways are uncertain. To prevent the model from improperly converging on incorrect predictions, we implement a strategy to maintain diversity among the pathways by deliberately selecting distinct masks. That is, if the maximum absolute occlusion score across all pathways at time $t$, $\max(\{|o_t^p|\}_{p=1}^P)$, is less than a predefined uncertainty threshold $\delta_{\text{conf}}$, we enforce the model to select mask candidates with unique IoU values. This is inspired by the observation that, within the same frame, different IoU scores typically correspond to distinct masks. In practice, we round each IoU score $\text{IoU}_t^{p,k}$ to two decimal places and only select those hypotheses with distinct rounded values.

Overall, the integration of constrained tree memory with uncertainty handling offers a balanced strategy that leverages multiple segmentation hypotheses to enhance robustness toward the long-term complex video and achieve more accurate and reliable segmentation performance by effectively mitigating error accumulation.

### 3.3 OBJECT-AWARE MEMORY BANK CONSTRUCTION

In each memory pathway, we devise object-aware memory selection to retrieve frames with discriminative objects. Meanwhile, we modulate the memory attention calculation to further strengthen the model's focus on the target objects.

**Memory Frame Selection.** To construct a memory bank that provides effective object cues, we selectively choose frames from previous time steps based on the predicted object presence and segmentation quality. Starting from the frame immediately before the current frame $t$, we iterate backward through the prior frames $i = \{t - 1, t - 2, \ldots, 1\}$ in sequence. For each frame $i$, we retrieve its predicted occlusion score $o_i$ and IoU score $\text{IoU}_i$ as reference. We include frame $i$ in the memory bank if it satisfies the following criteria:

$$\text{IoU}_i > \delta_{\text{IoU}} \quad \text{and} \quad o_i > 0,$$

where $\delta_{\text{IoU}}$ is a predefined IoU threshold. This ensures that only frames with confidently detected objects and reasonable segmentation masks contribute to the memory. We continue this process until we have selected up to $N$ frames. In contrast to SAM2, which directly picks the nearest $N$ frames as the memory entries, this selection process effectively filters out frames where the object may be occluded, absent, or poorly segmented, thereby providing more robust object cues for the segmentation of the current frame.

**Memory Attention Modulation.** To further emphasize more reliable memory entries during the cross-attention computation, we utilize the associated occlusion score $o_t$ to modulate the contribution of each memory entry. We define a set of standard weights $\mathcal{W}^{\text{std}}$ that are linearly spaced between a lower bound $w_{\text{low}}$ and an upper bound $w_{\text{high}}$:

$$\mathcal{W}^{\text{std}} = \left\{ w_{\text{low}} + \frac{i - 1}{N}(w_{\text{high}} - w_{\text{low}}) \right\}_{i=1}^{N+1}.$$

Next, we sort the occlusion scores in ascending order to obtain sorted indices $\mathcal{I}' = \{I_i\}_{i=1}^{N+1}$ such that:

$$o_{I_1} \leq o_{I_2} \leq \cdots \leq o_{I_{N+1}}.$$

We then assign the standard weights to the memory entries based on these sorted indices:

$$w_{I_i} = \mathcal{W}_i^{\text{std}}, \quad \text{for } i = 1, 2, \ldots, N + 1.$$

This assignment ensures that memory entries with higher occlusion scores, which indicate object presence with higher confidence, receive higher weights. Then, we linearly scale the original keys $\mathbf{M}_\tau$ with their corresponding weights:

$$\widetilde{\mathbf{M}}_\tau = w_\tau \cdot \mathbf{M}_\tau, \quad \text{for } \tau \in \mathcal{I}.$$

Finally, the modulated memory keys $\widetilde{\mathcal{M}}_t = \{\widetilde{\mathbf{M}}_\tau\}_{\tau \in \mathcal{I}}$ are used in the memory module's cross-attention mechanism to update the current frame's features. By using the available occlusion scores as indicators, we effectively emphasize memory entries with more reliable object cues while introducing minimal computational overhead.

## 4 EXPERIMENTS

### 4.1 DATASETS

To evaluate our method, we select 6 standard VOS benchmarks and report the following metrics: $\mathcal{J}$ (region similarity), $\mathcal{F}$ (contour accuracy), and the combined $\mathcal{J}\&\mathcal{F}$. All evaluations are conducted in a semi-supervised setting, where the first-frame mask is provided. The datasets used for testing are detailed as follows:

**SA-V** (Ravi et al., 2024) is a large-scale video segmentation dataset designed for promptable visual segmentation across diverse scenarios. It encompasses 50.9K video clips, aggregating to 642.6K masklets with 35.5M meticulously annotated masks. The dataset presents a challenge with its inclusion of small, occluded, and reappearing objects throughout the videos. The dataset is divided into training, validation, and testing sets, with most videos allocated to the training set for robust

model training. The validation set has 293 masklets across 155 videos for model tuning, while the testing set includes 278 masklets across 150 videos for comprehensive evaluation.

**LVOS v1** (Hong et al., 2023) is a VOS benchmark for long-term video object segmentation in realistic scenarios. It comprises 720 video clips with 296,401 frames and 407,945 annotations, with an average video duration of over 60 seconds. LVOS introduces challenging elements such as long-term object reappearance and cross-temporal similar objects. In LVOS v1, the dataset includes 120 videos for training, 50 for validation, and 50 for testing.

**LVOS v2** (Hong et al., 2024) expends LVOS v1 and provides 420 videos for training, 140 for validation, and 160 for testing. This paper primarily utilizes v2, as it already includes the sequences present in v1. The dataset spans 44 categories, capturing typical everyday scenarios, with 12 of these categories deliberately left unseen to evaluate and better assess the generalization capabilities of VOS models.

**Long Videos Dataset** (Liang et al., 2020) contains 3 long-form video sequences, each averaging over 2,000 frames, designed to evaluate VOS performance in real-world scenarios. For evaluation, 20 frames from each video are uniformly annotated.

**VOST** (Tokmakov et al., 2023) is a semi-supervised video object segmentation benchmark that emphasizes complex object transformations. Unlike other datasets, VOST includes objects that are broken, torn, or reshaped, significantly altering their appearance. It comprises more than 700 high-resolution videos, captured in diverse settings, with an average duration of 21 seconds, all densely labeled with instance masks.

**DAVIS2017** (Pont-Tuset et al., 2017) is a well-known benchmark dataset comprising 60 training videos and 30 validation videos, with a total of 6,298 frames. It offers high-quality, pixel-level annotations for every frame, making it a standard resource for evaluating different VOS methods.

### 4.2 MAIN RESULTS

**SAM2Long consistently improves SAM2 over all model sizes and datasets.** Table 1 presents an overall comparison between SAM2 and SAM2Long across various model sizes on the SA-V validation and test sets, as well as the LVOS v2 validation set. SAM2Long consistently outperforms the SAM2 baseline by a large margin. For instance, on the SA-V test set, SAM2Long-Small achieves a $\mathcal{J}\&\mathcal{F}$ score of 77.8, showing an improvement of 4.9 over SAM2-Small. Similarly, SAM2Long-Large achieves a $\mathcal{J}\&\mathcal{F}$ score of 80.3 on the SA-V test set, with a notable 4.7 improvement over SAM2-Large. This trend is also reflected in the LVOS validation set, where SAM2Long demonstrates considerable performance gains of at least 2.9 over SAM2 of the corresponding model size. These results showcase the effectiveness of the training-free memory tree in various video scenarios.

Table 1: Performance comparison on SA-V (Ravi et al., 2024) and LVOS v2 (Hong et al., 2024) datasets between SAM2 and SAM2Long across all model sizes. † We report the re-produced performance of SAM2 using its open-source code and checkpoint.

| Method | SA-V val | | | SA-V test | | | LVOS v2 val | | |
|---|---|---|---|---|---|---|---|---|---|
| | $\mathcal{J}\&\mathcal{F}$ | $\mathcal{J}$ | $\mathcal{F}$ | $\mathcal{J}\&\mathcal{F}$ | $\mathcal{J}$ | $\mathcal{F}$ | $\mathcal{J}\&\mathcal{F}$ | $\mathcal{J}$ | $\mathcal{F}$ |
| SAM2-Tiny† | 73.6 | 70.2 | 77.0 | 74.6 | 71.1 | 78.1 | 76.7 | 73.3 | 80.0 |
| SAM2Long-Tiny | 76.7 (3.1↑) | 73.0 | 80.5 | 78.6 (4.0↑) | 74.6 | 82.6 | 80.0 (3.3↑) | 76.5 | 83.5 |
| SAM2-Small† | 72.9 | 69.6 | 76.2 | 74.2 | 70.6 | 77.8 | 78.0 | 74.4 | 81.6 |
| SAM2Long-Small | 77.8 (4.9↑) | 74.0 | 81.6 | 77.9 (3.7↑) | 73.9 | 81.8 | 80.9 (2.9↑) | 77.3 | 84.6 |
| SAM2-Base+† | 75.3 | 71.9 | 78.7 | 74.8 | 71.3 | 78.2 | 77.3 | 73.9 | 80.6 |
| SAM2Long-Base+ | 79.3 (4.0↑) | 75.5 | 83.0 | 78.1 (3.3↑) | 74.3 | 81.8 | 80.5 (3.2↑) | 77.0 | 84.1 |
| SAM2-Large† | 76.6 | 73.3 | 79.8 | 75.6 | 72.3 | 79.0 | 79.3 | 74.6 | 84.1 |
| SAM2Long-Large | 81.3 (4.7↑) | 77.5 | 85.0 | 80.3 (4.7↑) | 76.4 | 84.2 | 83.5 (4.2↑) | 79.9 | 87.0 |

**SAM2Long outperforms previous methods and excels in unseen categories.** We also compare our proposed method, SAM2Long, with various state-of-the-art VOS methods on both the SA-V (Ravi et al., 2024) and LVOS (Hong et al., 2023; 2024) datasets, as shown in Table 2 and 3. Although SAM2 already surpasses previous methods by a large margin, SAM2Long pushes these limits even further. Specifically, our method achieves a $\mathcal{J}\&\mathcal{F}$ score of 81.3 on the SA-V validation set, a

5.2-point improvement over SAM2. For LVOS, SAM2Long respectively attains a $\mathcal{J}\&\mathcal{F}$ score of 82.3 and 83.5, outperforming SAM2 by 4.4 and 3.7 points on v1 and v2 subset. Notably, SAM2Long particularly excels in unseen categories, achieving $\mathcal{J}$ and $\mathcal{F}$ scores of 79.6 and 87.4. The significant improvements of 8.0 and 6.3 points over SAM2 highlight its robust generalization capabilities.

Table 2: Performance comparison with the-state-of-the-arts methods on SA-V dataset.

| Method | SA-V val | | | SA-V test | | |
|---|---|---|---|---|---|---|
| | $\mathcal{J}\&\mathcal{F}$ | $\mathcal{J}$ | $\mathcal{F}$ | $\mathcal{J}\&\mathcal{F}$ | $\mathcal{J}$ | $\mathcal{F}$ |
| STCN (Cheng et al., 2021) | 61.0 | 57.4 | 64.5 | 62.5 | 59.0 | 66.0 |
| RDE (Li et al., 2022) | 51.8 | 48.4 | 55.2 | 53.9 | 50.5 | 57.3 |
| SwinB-AOT (Yang et al., 2021a) | 51.1 | 46.4 | 55.7 | 50.3 | 46.0 | 54.6 |
| SwinB-DeAOT (Yang & Yang, 2022) | 61.4 | 56.6 | 66.2 | 61.8 | 57.2 | 66.3 |
| XMem (Cheng & Schwing, 2022) | 60.1 | 56.3 | 63.9 | 62.3 | 58.9 | 65.8 |
| DEVA (Cheng et al., 2023) | 55.4 | 51.5 | 59.2 | 56.2 | 52.4 | 60.1 |
| Cutie-base+ Cheng et al. (2024) | 61.3 | 58.3 | 64.4 | 62.8 | 59.8 | 65.8 |
| SAM2 (Ravi et al., 2024) | 76.1 | 72.9 | 79.2 | 76.0 | 72.6 | 79.3 |
| **SAM2Long (ours)** | 81.3 | 77.5 | 85.0 | 80.3 | 76.4 | 84.2 |

Table 3: Performance comparison with the-state-of-the-arts methods on validation set of LVOS dataset. Subscript $s$ and $u$ denote scores in seen and unseen categories.

| Method | LVOS v1 | | | LVOS v2 | | | | |
|---|---|---|---|---|---|---|---|---|
| | $\mathcal{J}\&\mathcal{F}$ | $\mathcal{J}$ | $\mathcal{F}$ | $\mathcal{J}\&\mathcal{F}$ | $\mathcal{J}_s$ | $\mathcal{F}_s$ | $\mathcal{J}_u$ | $\mathcal{F}_u$ |
| LWL (Bhat et al., 2020) | 56.4 | 51.8 | 60.9 | 60.6 | 58.0 | 64.3 | 57.2 | 62.9 |
| CFBI (Yang et al., 2020) | 51.5 | 46.2 | 56.7 | 55.0 | 52.9 | 59.2 | 51.7 | 56.2 |
| STCN (Cheng et al., 2021) | 48.9 | 43.9 | 54.0 | 60.6 | 57.2 | 64.0 | 57.5 | 63.8 |
| RDE (Li et al., 2022) | 53.7 | 48.3 | 59.2 | 62.2 | 56.7 | 64.1 | 60.8 | 67.2 |
| DeAOT (Yang et al., 2021a) | - | - | - | 63.9 | 61.5 | 69.0 | 58.4 | 66.6 |
| XMem (Cheng & Schwing, 2022) | 52.9 | 48.1 | 57.7 | 64.5 | 62.6 | 69.1 | 60.6 | 65.6 |
| DDMemory (Hong et al., 2023) | 60.7 | 55.0 | 66.3 | - | - | - | - | - |
| SAM2 (Ravi et al., 2024) | 77.9 | 73.1 | 82.7 | 79.8 | 80.0 | 86.6 | 71.6 | 81.1 |
| **SAM2Long (ours)** | 82.3 | 77.4 | 87.2 | 83.5 | 80.0 | 86.9 | 79.6 | 87.4 |

**SAM2Long demonstrates versatility when handling videos with various challenges.** In addition to the SA-V and LVOS datasets, we evaluate our proposed SAM2Long on other VOS benchmarks in Table 4. On the Long Videos Dataset (Liang et al., 2020), SAM2Long shows a significant performance boost over SAM2 by an improvement of 6.6 points. This validates the intuition that SAM2 faces challenges with error accumulation during long-term segmentation, while our SAM2Long with constrained memory tree effectively mitigates this issue. For the VOST dataset (Tokmakov et al., 2023), which focuses on complex object transformations, SAM2Long obtains a $\mathcal{J}\&\mathcal{F}$ score of 52.8, highlighting its capability to handle challenging transformations. On the DAVIS2017 dataset (Pont-Tuset et al., 2017), despite it being a relatively small, short-term benchmark where SAM2 already scores a high 89.8, SAM2Long still provides a 0.5-point gain. These results underscore the robustness and versatility of SAM2Long across a range of VOS benchmarks.

Table 4: The performance comparisons between SAM2 and SAM2Long on other VOS benchmarks. All experiments use SAM2-Large model. [†] We report the re-produced performance of SAM2 using its open-source code and checkpoint.

| Dataset | SAM2[†] | | | SAM2Long | | |
|---|---|---|---|---|---|---|
| | $\mathcal{J}\&\mathcal{F}$ | $\mathcal{J}$ | $\mathcal{F}$ | $\mathcal{J}\&\mathcal{F}$ | $\mathcal{J}$ | $\mathcal{F}$ |
| Long Videos Dataset (Liang et al., 2020) | 71.6 | 70.2 | 73.0 | 78.2 | 76.3 | 80.0 |
| VOST (Tokmakov et al., 2023) | 51.4 | 46.3 | 56.6 | 52.8 | 46.9 | 58.6 |
| DAVIS2017 (Pont-Tuset et al., 2017) | 89.8 | 87.1 | 92.5 | 90.3 | 87.6 | 93.0 |

## 4.3 ABLATION STUDY

We conduct a series of ablation studies on the validation split of SA-V dataset and use SAM2-Large as default model size.

Table 5: Ablation on number of pathways $P$.

| $P$ | $\mathcal{J}\&\mathcal{F}$ | $\mathcal{J}$ | $\mathcal{F}$ | Speed |
|---|---|---|---|---|
| 1 | 76.6 | 73.3 | 79.8 | $1\times$ |
| 2 | 80.9 | 77.3 | 84.5 | $0.93\times$ |
| 3 | 81.3 | 77.5 | 85.0 | $0.82\times$ |
| 4 | 81.2 | 77.5 | 84.9 | $0.75\times$ |

Table 6: Ablation on IoU threshold $\delta_{\text{iou}}$.

| $\delta_{\text{iou}}$ | $\mathcal{J}\&\mathcal{F}$ | $\mathcal{J}$ | $\mathcal{F}$ |
|---|---|---|---|
| 0 | 80.5 | 76.7 | 84.2 |
| 0.3 | 81.3 | 77.5 | 85.0 |
| 0.7 | 80.6 | 76.8 | 84.4 |
| 0.9 | 78.0 | 74.6 | 81.4 |

Table 7: Ablation on uncertainty threshold $\delta_{\text{conf}}$.

| $\delta_{\text{conf}}$ | $\mathcal{J}\&\mathcal{F}$ | $\mathcal{J}$ | $\mathcal{F}$ |
|---|---|---|---|
| 0.5 | 80.9 | 77.2 | 84.6 |
| 2 | 81.3 | 77.5 | 85.0 |
| 5 | 80.9 | 77.3 | 84.6 |

Table 8: Ablation on modulation $[w_{\text{low}}, w_{\text{high}}]$.

| $[w_{\text{low}}, w_{\text{high}}]$ | $\mathcal{J}\&\mathcal{F}$ | $\mathcal{J}$ | $\mathcal{F}$ |
|---|---|---|---|
| $[1, 1]$ | 80.7 | 77.0 | 84.3 |
| $[0.95, 1.05]$ | 81.3 | 77.5 | 85.0 |
| $[0.9, 1.1]$ | 81.0 | 77.4 | 84.7 |

**Number of Memory Pathways** $P$**.** We ablate the number of memory pathways to assess their impact on SAM2Long in Table 5. Note that setting $P = 1$ reverts to the SAM2 baseline. Increasing the number of memory pathways to $P = 2$ yields a notable improvement, raising the $\mathcal{J}\&\mathcal{F}$ score to 80.9. This result demonstrates that the proposed memory tree effectively boosts the model's ability to track the correct object while reducing the impact of occlusion. Further increasing the number of memory pathways to $P = 3$ achieves the best performance. However, using $P = 4$ shows no additional gains, suggesting that three pathways strike the optimal balance between accuracy and computational efficiency for the SAM2 model.

In terms of speed, since pruning is performed at every step, the speed is effectively maintained. Using three memory pathways slows down the model by only $18\%$, while yielding nearly a 5-point increase in performance.

**Iou Threshold** $\delta_{\text{iou}}$**.** The choice of the IoU threshold $\delta_{\text{iou}}$ is crucial for selecting frames with reliable object cues. As shown in Table 6, setting $\delta_{\text{iou}} = 0.3$ yields the highest $\mathcal{J}\&\mathcal{F}$, indicating an effective trade-off between filtering out poor-quality frames and retaining valuable segmentation information. In contrast, having no requirement on mask quality and feeding all masks containing objects into memory ($\delta_{\text{iou}} = 0$) decreases the score to 80.5, showing that unreliable frames with poor segmentation harm the SAM2 model. Meanwhile, an overly strict selection ($\delta_{\text{iou}} = 0.9$) degrades performance even more severely to 78.0, as it excludes too many potentially important neighboring frames, causing the model to rely on frames that are too far away from the current frame as memory.

**Uncertainty Threshold** $\delta_{\text{conf}}$**.** The uncertainty threshold $\delta_{\text{conf}}$ controls the selection of hypotheses under uncertain conditions. Our results in Table 7 indicate that setting $\delta_{\text{conf}}$ to 2 provides the highest $\mathcal{J}\&\mathcal{F}$ score, indicating an optimal level for uncertainty handling. Lower values (e.g., 0.5) result in suboptimal performance, as they may prematurely commit to incorrect segmentation hypotheses, leading to significant performance drops due to error propagation. On the other hand, higher values (e.g., 5) do not further improve performance, suggesting that beyond a certain threshold, the model does not benefit from additional mask diversity and can efficiently rely on the top-scoring masks when the segmentation is confident.

**Memory Attention Modulation** $[w_{\text{low}}, w_{\text{high}}]$**.** We explore the effect of modulating the attention weights for memory entries using different ranges in Table 8. The configuration $[1, 1]$ means no modulation is applied. We find that the configuration of $[0.95, 1.05]$ achieves the best performance while increasing the modulation range to ($[0.9, 1.1]$) slightly decreases performance. This result indicates that slight modulation sufficiently emphasizes reliable memory entries.

## 4.4 VISUALIZATION

We present a qualitative comparison between SAM2 and SAM2Long in Figure 3. SAM2Long demonstrates a significant reduction in segmentation errors, maintaining more accurate and consistent tracking of objects across various frames.

For example, in the second sequence of the first row, SAM2 immediately loses track of the man of interest when occlusion happens. Although SAM2Long also loses track initially, its memory

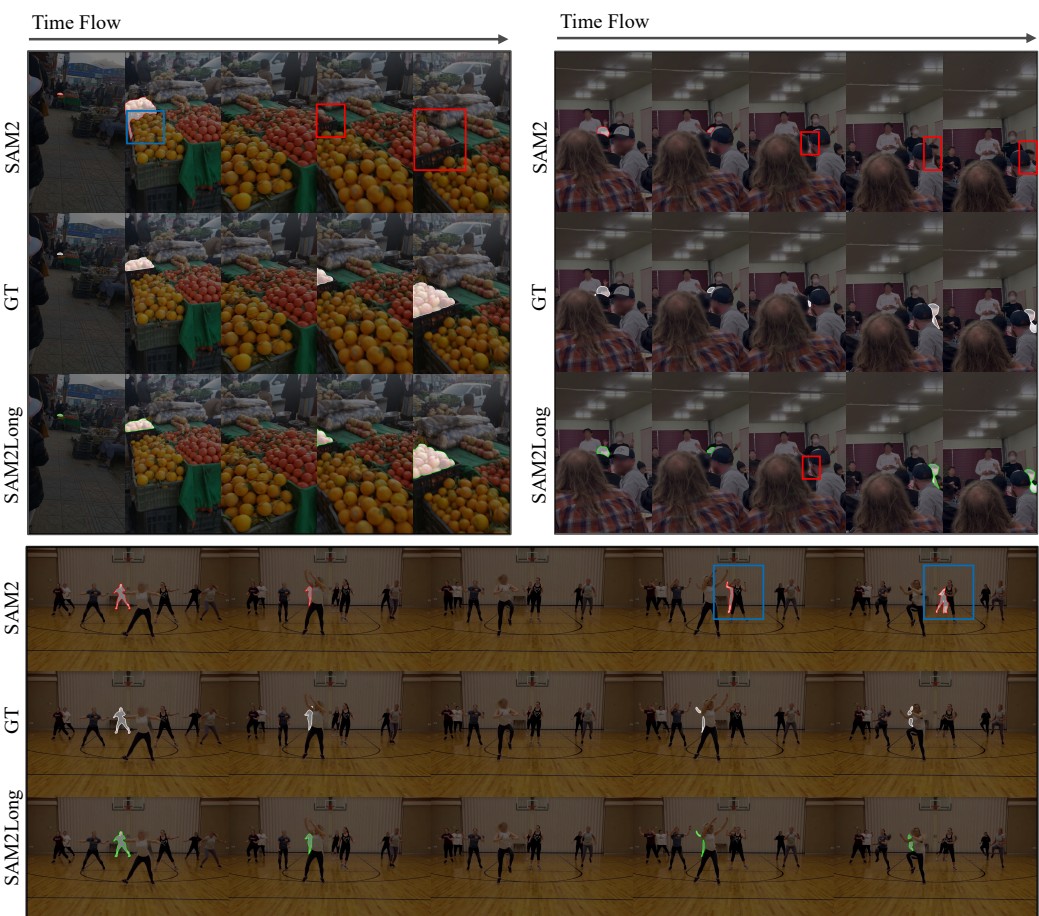

Figure 3: Qualitative comparison between SAM2 and SAM2Long, with GT (Ground Truth) provided for reference. A blue box is used to highlight incorrectly segmented objects, while a red box indicates missing objects. Best viewed when zoomed in.

tree with multiple pathways enables it to successfully re-track the correct man later on. In another case, depicted in the second row where a group of people is dancing, SAM2 initially tracks the correct person. However, when occlusion occurs, SAM2 mistakenly switches to tracking a different, misleading individual. In contrast, SAM2Long handles this ambiguity effectively. Even during the occlusion, SAM2Long manages to resist the tracking error and correctly resumes tracking the original dancer when she reappears.

In conclusion, SAM2Long significantly improves SAM2's ability to handle object occlusion and reappearance, thereby enhancing its performance in long-term video segmentation.

## 5 CONCLUSION

In this paper, we introduce SAM2Long, a training-free enhancement to SAM2 that alleviates its limitations in long-term video object segmentation. By employing a constrained tree memory structure with object-aware memory modulation, SAM2Long effectively mitigates error accumulation and improves robustness against occlusions, resulting in a more reliable segmentation process over extended periods. Extensive evaluations on six VOS benchmarks demonstrate that SAM2Long consistently outperforms SAM2, especially in complex video scenarios. Notably, SAM2Long achieves up to a 5-point improvement in $J\&F$ scores on challenging long-term video benchmarks such SA-V and LVOS without requiring additional training or external parameters.

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
