# OpenReview forum: "SAM2Long: Enhancing SAM2 for Long Video Segmentation with a Training-Free Memory Tree"
_ICLR.cc/2025/Conference — ICLR 2025 Conference Withdrawn Submission_

### Official Review · Reviewer_8TNc · 2024-10-28

**Soundness:** 2
**Presentation:** 2
**Contribution:** 2
**Rating:** 5
**Confidence:** 4

**Summary:**

The paper proposes SAM2Long, a method comprising multiple strategies to enhance SAM2. SAM2Long maintains multiple memory banks and selectively includes frames with high-quality segmentation masks to reduce error accumulation. The method demonstrates improved performance across multiple benchmarks.

**Strengths:**

1. The paper is easy to understand.

2. Constructing multiple memory banks to reduce error accumulation is reasonable, and the method achieves improved performance on VOS.

**Weaknesses:**

1. This paper expands memory banks to improve performance, the idea is reasonable but brings limited new insights.

2. The method comprises a set of strategies with hyperparameters, adding complexity to the baseline and affecting efficiency, with unclear generalization.

3. The ablation study is incomplete, making it difficult to understand the effect of each individual strategy.

4. The limitations of the proposed method are not discussed.

**Questions:**

See Weaknesses.

---

> ### Author Response · Authors · 2024-11-14
>
> We sincerely thank the reviewer for your insightful comments and recognition of this work, especially for acknowledging that:
>
> 1. The paper is easy to follow.
> 2. The approach of constructing multiple memory banks to mitigate error accumulation is logical, and it leads to improved performance in VOS.
>
>
> Thanks again for your very constructive comments, which have helped us improve the paper quality significantly! Below we would like to provide point-to-point responses to all the raised questions and weaknesses:
>
> >**Q1: Limited new insights.**
>
> **A1**:
> We respectfully disagree with this point. While improving the memory bank is indeed a key aspect of video tasks, the specific enhancement of the memory module in the state-of-the-art video model SAM 2 has not been previously explored. Our work is the first to propose an improvement to the memory mechanism in SAM 2. We introduce a tree structure to search at the video level during inference time, with the insight that such test-time computation can significantly enhance the performance of SAM 2.
>
> >**Q2: Complexity and generalization.**.
>
> **A2**:
> In terms of complexity, the increase in FLOPS compared to the baseline SAM2 is minimal, as detailed in the table below. We decompose the computation process into four key modules and report the results for SAM2-Large as a reference:
>
> | Number of pathways $p$|1 | 2| 3|
> |------- | -----|-----|------- |
> |Image Encoder | 810 GFlops |810 GFlops |810 GFlops |
> |Memory Encoder| 5 GFlops | 5x2 GFlops | 5x3 GFlops|
> |Mask Decoder | 1.7 GFlops | 1.7x2 GFlops | 1.7x3 GFlops |
> |Memory Attention | 27.4  GFlops | 27.4x2  GFlops | 27.4x3  GFlops|
> |Total GFlops | 844.1 (1x) | 878.2 (1.04x) | 912.3 (1.08x)|
> |Throughput | 1x  | 0.93x | 0.82x |
> |Parameter| 224M|224M|224M|
> |J&F | 76.6 | 80.9 | 81.3 |
>
> As shown, the increase in FLOPS is modest across different pathway configurations, and the parameter count remains unchanged.
>
> Regarding generalization, our method has been rigorously tested across multiple benchmarks. We evaluated our approach on four different model sizes of SAM2 across six benchmarks. Notably, for short-term video datasets, our method shows no negative impact on performance. However, for long-term video object segmentation (VOS) benchmarks, such as SA-V and LVOS, we observe significant and consistent improvements.
>
>
> >**Q3: Incomplete ablation study**
>
> **A3**:
> Thank you for your feedback. Could you please clarify which specific part of the strategy has not been verified? We are happy to supplement the ablation study in a later version based on your suggestions.
>
>
> >**Q4: Limitations**
>
> **A4**:
> While our method shows promise, its performance is ultimately constrained by the capacity of SAM2, as we do not modify any learnable parameters within SAM2. This limitation presents an opportunity for future improvements. Additionally, we have not yet addressed multi-object scenarios. Exploring the semantic interactions between multiple objects within the same frame could provide valuable insights for achieving more accurate segmentation.
>
> ---
> Please don’t hesitate to let us know if there are any additional clarifications or experiments that we can offer!

---

### Official Review · Reviewer_F3HN · 2024-11-01

**Soundness:** 3
**Presentation:** 3
**Contribution:** 2
**Rating:** 3
**Confidence:** 5

**Summary:**

The paper focuses on tackling the problem of video object segmentation for long videos. The proposed solution, SAM2Long, is an enhancement to the Segment Anything Model 2 (SAM2), designed to address its limitations in segmenting long videos, specifically in handling occlusions and preventing error accumulation. To achieve this, SAM2Long utilizes a training-free, constrained memory tree structure that tracks multiple segmentation pathways, selecting the most reliable ones over time. The effectiveness of the method is demonstrated through evaluations across various benchmarks, showing significant performance improvements over SAM2 in terms of J&F scores on occlusion-heavy and long-term video object segmentation datasets.

**Strengths:**

1. The proposed approach is intuitively reasonable.

2. The proposed SAM2Long outperforms strong SAM2 on standard VOS benchmarks without extra training.

3. The proposed approach does not significantly raise computational expenses.

**Weaknesses:**

1. The paper provides little insights for the video object segmentation community. Although the constrained memory tree and memory bank management strategies are useful, they seem to be heuristic engineering rather than research findings. For instance, all mask selection processes are treated on a case-by-case basis. Consequently, it seems unlikely that the approach can handle all possible scenarios. Does SAM2 consistently predict the IoU/occlusion scores for both small and large instances? How can we ensure the reliability of the predicted IoU/occlusion scores? What happens if SAM2 produces incorrect predictions for all N candidates? Are there no other cues available to enhance the robustness of VOS beyond IoU/occlusion scores?

2. The proposed method restricts SAM2 to ground-truth mask prompts. One of the strengths of SAM2 lies in its ability to handle various types of prompts (e.g., boxes, points, masks) and interactive scenarios. However, the presented approach eliminates this versatility by overfitting to a single scenario.

3. The paper lacks discussions on limitations or analyses of failure cases. Simply presenting a single failure case in the supplementary video without further analysis offers limited value to the readers.

4. There is no analysis of the effectiveness in long-term and occlusion-heavy scenarios. The paper only presents quantitative and qualitative results on full benchmark datasets. As a result, it is difficult to conclude that this work truly excels in such challenging conditions, as argued in L107. The paper can provide a performance comparison based on the number of frames and the number of occluded instances.

5. The paper reports lower performance of the SAM2 baselines compared to the official implementation (see https://github.com/facebookresearch/sam2#model-description). The paper should explain the reasons behind this performance gap.

6. Experimental results on YouTube-VOS 2019 are missing. This benchmark is widely used in the VOS field, and SAM2 demonstrates strong performance on it. Given that YouTube-VOS 2019 also includes complex occlusion-heavy scenes, it would be beneficial to include the results from this benchmark in the paper.

7. L162: we redesign -> We redesign

**Questions:**

My primary concerns include the limited insights, overfitting to the mask prompt, and a lack of experimental analysis. My initial recommendation stands at rejection; however, I'm open to revising this rating if the aforementioned shortcomings are properly addressed.

**Details Of Ethics Concerns:**

No ethics concerns.

---

> ### Author Response · Authors · 2024-11-14
>
> We sincerely thank the reviewer for your insightful comments and recognition of this work, especially for acknowledging that:
>
> 1. The proposed approach is conceptually sound.
>
> 2. SAM2Long demonstrates superior performance over the strong SAM 2 on standard VOS benchmarks, without requiring additional training.
>
> 3. The approach does not lead to a significant increase in computational costs.
>
> Thanks again for your very constructive comments, which have helped us improve the paper quality significantly! Below we would like to provide point-to-point responses to all the raised questions and weaknesses:
>
> >**Q1: Limited insights and readability of the method.**.
>
> **A1**:
> We respectfully disagree with the reviewer’s assessment that our contribution lacks significant insights for the video object segmentation (VOS) community.
>
> Our contribution is the novel design of a constrained tree memory structure, introduced during test-time inference rather than at training. This enhancement allows SAM 2 to dynamically explore multiple memory pathways in a resource-efficient manner, mitigating error propagation and improving robustness over long sequences.
>
> Although heuristic methods have been applied in some contexts, the idea of leveraging test-time search via a memory tree to address challenges like error accumulation and long-term object tracking is unique to our work. The insight that this structure can significantly improve performance without additional training or external parameters adds value to the VOS community. Furthermore, our ability to manage memory diversity through occlusion-aware pruning and selective memory updates offers an advantage in handling difficult video scenarios—an area not addressed in prior SAM 2 work.
>
> Thus, we believe our approach represents a meaningful research-driven enhancement to SAM 2’s memory system, contributing both theoretically and practically by addressing the limitations of existing methods for complex, long-term video object segmentation.
>
> Regarding reliability, we acknowledge that SAM2Long inherits limitations from SAM 2, but it is important to recognize that no model can address every possible edge case. However, it is important to highlight that SAM 2 currently outperforms all prior methods on the SA-V and LVOS benchmarks. Our approach pushes these boundaries even further, yielding an average improvement of 3.8 points. This advancement is a clear demonstration of how we have enhanced the state-of-the-art model, representing a significant contribution to the field.
>
> >**Q2: Overfitting to mask prompt.**
>
>
> **A2**: We respectfully disagree with the reviewer’s assessment. It seems there may be a misunderstanding of our method. The prompts (e.g., boxes, points, masks) and interactive mode mentioned by the reviewer are orthogonal to and independent of our contribution. Specifically, our approach does not alter the nature of how different types of prompts are processed. Whether they are point-based interactions or box-based inputs, all prompts first pass through a prompt encoder and mask decoder to generate masks. These masks are then stored in a memory bank.
>
> Our contribution primarily focuses on improving the memory mechanism, which is compatible with a variety of prompt types, as seen in the original SAM2. The memory mechanism enables the model to retain predicted masks for more efficient and effective segmentation, allowing for enhanced performance when revisiting previous predictions. Our approach does not limit SAM2's versatility but instead enhances the handling of memory, which in turn improves the segmentation process.
>
>
>
> >**Q3: Limitations and Analysis of Failure Cases.**
>
>
> **A3**:
> While our method demonstrates significant promise, its performance is inherently constrained by the capacity of SAM 2, as we do not modify any learnable parameters within SAM 2. This limitation provides an opportunity for future improvements. For instance, the model may still track the wrong object during segmentation if it remains occluded, as the uncertainty handling encourages storing all possible candidates in the memory frame selection. This introduces a trade-off between exploring potential solutions and avoiding incorrect associations, which can be addressed in future work to further enhance robustness.
>
>
> >**Q4: No Analysis on long-term and occlusion-heavy scenarios.**
>
> **A4**: We encourage to take a look at Figure 1(b). There is a plot showing that our method has greater resilience to elapsed time compared to SAM 2, maintaining superior performance over longer periods.

---

> > ### Author Response · Authors · 2024-11-14
> >
> > >**Q5: Lower baseline to the official implementation.**
> >
> > **A5**: We provide a detailed comparison in the table below, showing official numbers, our reproduced results, and SAM2Long. As you can see, we do not always show lower performance. In fact, even compared to the official numbers, our method demonstrates significant improvement.
> > | SAM 2 | SA-V test | LVOS v2 |
> > |----|-----|-------|
> > | Official-T | 75.0 | 75.3 |
> > | Reproduce-T | 74.6 | 76.7 |
> > | Ours-T | 76.7 | 80.0 |
> > | Official-S | 74.9 | 76.4  |
> > | Reproduce-S | 74.2 |78.0 |
> > | Ours-S | 74.2 | 80.9 |
> > | Official-B | 74.7 | 75.8 |
> > | Reproduce-B | 74.2 | 77.3 |
> > | Ours-B | 74.8 |80.5 |
> > | Official-L | 76.0 | 79.8 |
> > | Reproduce-L | 75.6 | 79.3 |
> > | Ours-L | 80.3 | 83.5 |
> >
> > >**Q6: Results on YouTube-VOS 2019**
> >
> > **A6**:
> > We did not include YouTube-VOS in our evaluation because it is a very short dataset, with an average duration of only 5 seconds. In such short time frames, it is not possible to demonstrate the long-term capabilities of our method. Objects may reappear within a very brief window, making it difficult to differentiate our method from SAM 2, as both models would perform similarly in these scenarios. In specific, SAM2Long and SAM 2 all achieve 88.7 on YouTube-VOS.

---

### Official Review · Reviewer_wca8 · 2024-11-02

**Soundness:** 3
**Presentation:** 4
**Contribution:** 3
**Rating:** 8
**Confidence:** 4

**Summary:**

Authors propose SAM2Long model that builds on the precursor SAM2 for video object segmentation and tracking while ensuring it works in long-term tracking scenarios. They propose using a constrained tree based memory within SAM2 framework that allows for exploring multiple valid pathways and overcoming issues with occlusion or going out of the field of view then back in. They evaluate on various benchmarks including LVOS, VOST in addition to standard benchmarks such as DAVIS showing consistent improvement over their baseline SAM2 with a considerable margin.

**Strengths:**

- Strong results and gain w.r.t the baseline SAM2
- Interesting work that focuses on segmentation and tracking in long videos which is indeed a challenging scenario.

**Weaknesses:**

- a full evaluation of the computational efficiency in terms of runtime, FLOPS and parameters is needed so we can evaluate the overhead of their method w.r.t SAM2 beyond what is presented in Table 5.

**Questions:**

- Ablation on the modulation in Table 8, how were these parameters ranges selected during the ablation? What happens outside that range? Also the changes seem to be quite minor and negligible in the last two rows so would be good to see what happens as we increase this?

---

> ### Author Response · Authors · 2024-11-14
>
> We sincerely thank the reviewer for your insightful comments and recognition of this work, especially for acknowledging that:
>
> 1. SAM2Long achieves strong results compared to baseline SAM 2.
> 2. An intriguing approach that addresses the challenges of segmentation and tracking in long videos.
>
> Thanks again for your very constructive comments, which have helped us improve the paper quality significantly! Below we would like to provide point-to-point responses to all the raised questions and weaknesses:
>
> >**Q1: Full evaluation of the computational efficiency**.
>
> **A1**:
> Thank you for your feedback. The number of parameters remains identical to SAM2 since no new parameters are introduced.
>
> For FLOPS, we decompose the computation process into four key modules and report the results for SAM2-Large as a reference:
>
> | Number of pathways $p$|1 | 2| 3|
> |------- | -----|-----|------- |
> |Image Encoder | 810 GFlops |810 GFlops |810 GFlops |
> |Memory Encoder| 5 GFlops | 5x2 GFlops | 5x3 GFlops|
> |Mask Decoder | 1.7 GFlops | 1.7x2 GFlops | 1.7x3 GFlops |
> |Memory Attention | 27.4  GFlops | 27.4x2  GFlops | 27.4x3  GFlops|
> |Total GFlops | 844.1 (1x) | 878.2 (1.04x) | 912.3 (1.08x)|
> |Throughput | 1x  | 0.93x | 0.82x |
> |Parameter| 224M|224M|224M|
> |J&F | 76.6 | 80.9 | 81.3 |
>
> This detailed evaluation demonstrates that the additional overhead introduced by our method is minimal in terms of computational cost, while still yielding improved performance.
>
> >**Q2: Ablation on Modulation**.
>
> **A2**:
> We intentionally did not modulate the features significantly, as we found that larger ranges, such as [0.85, 1.15], lead to a noticeable performance degradation, as shown in the table. We hypothesize that this is due to the dot-product attention mechanism, which is followed by a softmax operation. This softmax step amplifies the effects of any modulation, causing instability when the range is too large. As a result, we remain cautious in tuning the memory features to avoid performance loss.
>
> |Modulation Range [$w_\text{low}, w_\text{high}$]| J&F |
> |------- | ----------|
> |[0.95, 1.05]| 81.3 |
> |[0.9, 1.1]| 81.0 |
> |[0.85, 1.15] | 80.0 |
>
>
> ---
> Please don’t hesitate to let us know if there are any additional clarifications or experiments that we can offer!

---

### Official Review · Reviewer_QmER · 2024-11-03

**Soundness:** 3
**Presentation:** 3
**Contribution:** 2
**Rating:** 5
**Confidence:** 4

**Summary:**

This article presents SAM2Long, an improved training-free video object segmentation strategy designed to address the limitations of the Segment Anything Model 2 (SAM2) in long video segmentation. It identifies the "error accumulation" issue in SAM2's greedy memory selection design, where errors in mask predictions for one frame negatively impact subsequent frames, particularly in complex long videos. SAM2Long introduces a constrained tree memory structure that maintains a fixed number of segmentation pathways, allowing for the generation of multiple candidate branches for each frame and selecting the best pathways based on cumulative scores. To enhance robustness, it prevents premature convergence on incorrect predictions by selecting hypotheses with distinct predicted masks in uncertain situations and includes only high-quality segmentation masks and confidently detected objects in the memory bank. Empirical evaluations show that SAM2Long significantly outperforms SAM2 across various video object segmentation benchmarks, achieving an average improvement of 3.8 points, with some cases showing up to a 5-point increase in J&F scores on long-term video benchmarks. Overall, SAM2Long enhances the memory module and introduces a new selection mechanism, effectively improving segmentation and tracking of objects in complex long video scenarios.

**Strengths:**

1.The introduction of a constrained tree memory structure is a novel contribution that effectively addresses the limitations of the existing SAM2 model, particularly in handling long video segmentation tasks.
2.The method's ability to prevent error accumulation by maintaining multiple segmentation pathways enhances its robustness, which is crucial for real-world applications where occlusions and complex scenes are common.

**Weaknesses:**

This article introduces a new constrained tree memory structure to avoid error accumulation. The method compares the results obtained from multiple decodings, selecting the optimal one from each path to update the new memory bank. However, there are some unclear aspects in the description of the method:

1. Unclear Source of Memory Bank Updates: What is the source of the memory bank updates for each path? Is it derived from the previous state of another branch or the previous state of the current branch? The description of the tree memory structure is not clear and complete. It is recommended to provide a detailed explanation of this process or to include visual examples to help readers better understand.

2. Mask Storage and Calculation Method: Does each memory bank store the overall mask for all targets, or does it store individual masks for each target? The calculation of the IoU for the masks also needs clarification—does it represent the average of the masks for multiple targets, or is it calculated in another way? Additionally, are all the threshold settings designed based on statistical results, or are they set randomly? This point requires further explanation.

There are also shortcomings in the experimental results:

1. Lack of Comparative Experiments: There is a lack of comparative experiments with SAM2 on the YouTube VOS and MOSE datasets. These comparative experiments are crucial for validating the effectiveness of the new method, and it is suggested to supplement relevant results.

2. Unreasonable Parameter Design in Ablation Experiments: In Tables 6, 7, and 8, the threshold for conf is set to [0.5, 2, 5], which has too large of an interval and does not meet statistical standards. Furthermore, have all the thresholds been statistically analyzed based on the actual IoU and conf situations across all frames? Would it be better to design thresholds based on statistical data? Is there any inconsistency in the actual IoU and conf situations across different datasets? These questions need further exploration and explanation.

**Questions:**

1.Unclear Source of Memory Bank Updates: What is the source of the memory bank updates for each path? Is it derived from the previous state of another branch or the previous state of the current branch? The description of the tree memory structure is not clear and complete. It is recommended to provide a detailed explanation of this process or to include visual examples to help readers better understand.

2.Mask Storage and Calculation Method: Does each memory bank store the overall mask for all targets, or does it store individual masks for each target? The calculation of the IoU for the masks also needs clarification—does it represent the average of the masks for multiple targets, or is it calculated in another way? Additionally, are all the threshold settings designed based on statistical results, or are they set randomly? This point requires further explanation.

3.Lack of Comparative Experiments: There is a lack of comparative experiments with SAM2 on the YouTube VOS and MOSE datasets. These comparative experiments are crucial for validating the effectiveness of the new method, and it is suggested to supplement relevant results.

4.Unreasonable Parameter Design in Ablation Experiments: In Tables 6, 7, and 8, the threshold for conf is set to [0.5, 2, 5], which has too large of an interval and does not meet statistical standards. Furthermore, have all the thresholds been statistically analyzed based on the actual IoU and conf situations across all frames? Would it be better to design thresholds based on statistical data? Is there any inconsistency in the actual IoU and conf situations across different datasets? These questions need further exploration and explanation.

---

> ### Author Response · Authors · 2024-11-14
>
> We sincerely thank the reviewer for your insightful comments and recognition of this work, especially for acknowledging that:
>
> 1. The structure of constrained tree memory is a novel contribution.
> 2. The method is robust to real-world application.
>
> Thanks again for your very constructive comments, which have helped us improve the paper quality significantly! Below we would like to provide point-to-point responses to all the raised weaknesses:
>
> >**Q1: Unclear Source of Memory Bank Updates**.
>
>
> **A1**: The memory bank updates depend on the cumulative score from each branch. If the cumulative scores from a single branch are the highest among the available options, subsequent frames will continue using the memory from that branch. As illustrated in Figure 2, in the first frame update, both memory banks remain unchanged because the scores are all equal at 1.6. However, in later frames, memory bank 2 accumulates two higher mask scores (2.3 and 2.4). Consequently, we use the memory masks from this branch (shown in pink) as the memory for the current frame. As a result, we replace the original memory bank 1 (blue) with memory bank 2 (pink).
>
> >**Q2: Mask Storage and Calculation Method**.
>
> **A2**: We handle memory storage separately for each target. Each target maintains its own individual memory bank. All threshold values are empirically tuned based on performance on the SA-V dataset. We observed minimal impact on performance with slight variations in threshold values. For instance, as shown in the full version of Table 6, there is little change in performance (measured by J&F) as $\delta_{\text{iou}}$ varies from 0 to 0.7:
>
> | $\delta_{\text{iou}}$ |0.1|0.2|0.3|0.4|0.5|0.6|0.7|
> |-------|-----------|-----------|------|-------|-----|--------|-----------|
> | J&F  |   80.5| 80.8| 81.1| 81.3| 81.2| 81.0| 80.8| 80.6|
>
> >**Q3: Lack of Comparative Experiments**.
>
> **A3**: We did not include these datasets because they are relatively short in duration. For instance, MOSE has an average video length of only 12 seconds, and YouTube-VOS is even shorter. These shorter datasets make it challenging to demonstrate the long-term capability of our method. However, based on the new checkpoints released by SAM 2.1, we have evaluated the performance and report the results below:
>
> |Dataset|SAM 2.1 | SAM2.1Long|
> |------- | ----------|------- |
> |MOSE| 74.5 | 75.2 |
> |YouTubeVOS| 88.7 | 88.7 |
>
> Our results show no performance drop on these datasets; in fact, we observe a slight improvement on the MOSE benchmark. This indicates that our method does not negatively impact performance on any type of video and is especially effective for long-term settings.
>
>
> >**Q4: Unreasonable Parameter Design in Ablation Experiments**.
>
>
> **A4**: Due to page limits, we couldn’t include the full table and instead chose a few representative results to convey key insights.
> Thank you for the great question. While we could achieve better results by performing a grid search to optimize parameters for each specific dataset, this approach is not practical for real-world applications. Therefore, we did not focus on finding an *optimal* result for any particular dataset; our design aims to show general trends. For instance, setting the confidence threshold too high would result in fewer frames being stored in memory, causing these frames to be too distant to effectively support predictions for the current frame. In our findings, IoU prediction and confidence are closely aligned with ground truth, demonstrating robust generalization due to the extensive training dataset included in SAM 2.
>
>
>
>
> ---
> Please don’t hesitate to let us know if there are any additional clarifications or experiments that we can offer!

---

### Note · Authors · 2024-11-15

I have read and agree with the venue's withdrawal policy on behalf of myself and my co-authors.